# Contrastive Entity Linkage: Mining Variational Attributes from Large Catalogs for Entity Linkage

**Varun Embar**                                                VEMBAR@UCSC.EDU
*Univeristy of California, Santa Cruz*

**Bunyamin Sisman**                                          BUNYAMIS@AMAZON.COM
**Hao Wei**                                                      WEHAO@AMAZON.COM
**Xin Luna Dong**                                            LUNADONG@AMAZON.COM
*Amazon.com, Inc*

**Christos Faloutsos**                                       CHRISTOS@CS.CMU.EDU
*Carnegie Mellon University*

**Lise Getoor**                                              GETOOR@SOE.UCSC.EDU
*Univeristy of California, Santa Cruz*

## Abstract

Presence of near identical, but distinct, entities called *entity variations* makes the task of data integration challenging. For example, in the domain of grocery products, variations share the same value for attributes such as brand, manufacturer and product line, but differ in other attributes, called *variational attributes*, such as package size and color. Identifying variations across data sources is an important task in itself and is crucial for identifying duplicates. However, this task is challenging as the variational attributes are often present as a part of unstructured text and are domain dependent. In this work, we propose our approach, **Contrastive entity linkage**, to identify both entity pairs that are the same and pairs that are variations of each other. We propose a novel unsupervised approach, **VarSpot**, to mine domain-dependent variational attributes present in unstructured text. The proposed approach reasons about both similarities and differences between entities and can easily scale to large sources containing millions of entities. We show the generality of our approach by performing experimental evaluation on three different domains. Our approach significantly outperforms state-of-the-art learning-based and rule-based entity linkage systems by up to 4% F1 score when identifying duplicates, and up to 41% when identifying entity variations.

## 1. Introduction

Are the two tablets "TabMaker Color HD 16 GB" and "TabMaker Color HD 8 GB" the *same* or *different*? It is unclear – while they are essentially the same product – they have the same manufacturer and brand, they also have an important difference, storage size. In domains such as products and music, the presence of large numbers of nearly identical but distinct entities makes entity linkage challenging. We refer to these entities as *entity variations*, and in fact, whether these entities are distinct or not may even be context dependent. More formally, entity variations are sets of entities that share the same value

across core attributes called *base attributes*, but differ from each other along a few crucial attributes which we refer to as *variational attributes*. For example, variations of the tablet in the motivating example have different values for the variational attribute storage size.

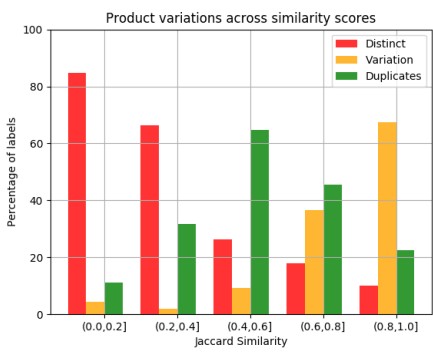

Figure 1: **Variation histogram:** The distribution of grocery product pairs which are duplicates, variations and distinct across Jaccard similarity computed on title. Even for high similarity (between $(\mathbf{0.8}, \mathbf{1.0}]$), about 70% of the pairs are actually variations.

An entity linkage framework for domains with variations needs to identify both entities that are exactly the same (duplicates), and also entity variations (variations). While identifying variations is important for matching duplicate entities, it is a useful task in its own right. For example, it is critical for enhancing the customer shopping experience. Consolidating product variations on the same page enables customers to locate the product they desire quickly. Grouping product variations in search results can increase diversity and prevent returned results from being dominated by multiple variants of the same product.

To illustrate the challenge of performing entity linkage in presence of variations, consider a sampling of grocery products from two e-commerce catalogs (details described in Section 5). These catalogs contain two attributes, brand and title. Fig. 1 shows the distribution of duplicates, variations and distinct entity pairs sampled and grouped into five uniform interval buckets using Jaccard similarity computed on the title. Across all ranges, variations are mixed in with duplicates and distinct pairs, with some buckets containing up to 70% variation pairs. There are no structured attributes to help distinguish them. As we show in our experiments, even state-of-the-art approaches that compute multiple similarity measures incorrectly link variations as duplicates.

To address these challenges, in this paper we propose a novel framework, **contrastive entity linkage (CEL)**, that identifies both duplicates and variations together.

|  | Duplicate Matching | Variation Matching | Variational Attrib. Extraction |
|---|---|---|---|
| ER approaches | ✓ | | |
| Li et al. [2015] | | ✓ | |
| Attribute extraction | | | ✓ |
| CEL | ✓ | ✓ | ✓ |

Table 1: CEL extracts variational attributes and identifies both duplicates and variations.

The main contributions of our work include:

**Three-way linkage:** We extend the traditional task of two class entity linkage, where the goal is to identify duplicates, to a three class setting, where along with duplicates we also identify entity variations.

**Automatic variational attribute discovery:** We propose a scalable, unsupervised variational attribute discovery approach, **VarSpot**. **VarSpot** analyzes similarities and differences between entities within the *same* catalog and uses the notion of *contrast features* to model variational attributes.

**Effectiveness:** We perform empirical evaluation in three different domains to show the

generality of our approach. Using three different state-of-the-art entity linkage frameworks, including rule-based and deep learning based frameworks, we show that models with contrast features significantly outperform models without them when identifying duplicates and variations. Further, through annotations using Mechanical Turk, we show the interpretable nature of contrast features.

## 2. Related Work

There is a wide body of research, spanning several decades, on the task of entity linkage [Ananthakrishna et al., 2002, Dong et al., 2005, Bhattacharya and Getoor, 2007, Dong et al., 2009, Sarawagi and Bhamidipaty, 2002, Domingos, 2004, Mudgal et al., 2018, Trivedi et al., 2018]. See [Elmagarmid et al., 2007, Köpcke and Rahm, 2010, Doan and Halevy, 2005, Koudas et al., 2006, Getoor and Machanavajjhala, 2012, Christen, 2012] for survey papers comparing various approaches. Starting from the seminal paper by Fellugi and Sunter [Fellegi and Sunter, 1969], several rule-based approaches [Fan et al., 2009, Singh et al., 2017], learning approaches [Bilenko et al., 2006, Konda et al., 2016, Singla and Domingos, 2006], crowdsourcing approaches [Gokhale et al., 2014, Wang et al., 2012, Stonebraker et al., 2013] and graph-based approaches [Zhu et al., 2016] have been proposed. Recently, approaches that use deep learning models have been proposed [Mudgal et al., 2018, Ebraheem et al., 2017]. Here we review recent work in entity linkage on products, followed by approaches that identify variations of entities. Finally, we review the recent work for task of attribute value extraction from text.

**Entity linkage for products:** The task of entity linkage for products has received a fair bit of attention recently [Kannan et al., 2011, Köpcke et al., 2012, Horch et al., 2016, Londhe et al., 2014]. Supervised approaches such as [Kannan et al., 2011, Köpcke et al., 2012, Ristoski et al., 2017] extract product attributes from the product title and description and use these extracted attributes to perform entity linkage. Kannan et al. [2011] use an inverted index to extract the set of product attributes from the product title and then use logistic regression to learn their importance. Köpcke et al. [2012] use regular expressions to extract attributes and use search engines to refine them. They train a SVM using the extracted attributes to link offers. Ristoski et al. [2017] use a convolutional neural network and a CRF to extract product features and train various supervised models such as random forests and SVM. Approaches such as Horch et al. [2016] learn a new similarity score by combining several distance scores such as Jaccard and Sorensen distance. Londhe et al. [2014] propose an unsupervised approach to link products using a community detection algorithm. They make use of a search engine to enrich the text of products. While these approaches identify duplicates, they do not identify variations.

**Variation Clustering:** Recasens et al. [2011] provide a theoretical framework for incorporating near-identity relations while performing entity linkange. Approaches such as Li et al. [2015] identify records in a catalog that are variations of each other. They take as input a catalog and a partitioning of the record attributes into common-valued, dominant-valued and multivalued attributes. Using a small set of labeled data, they learn weights for the attribute values. Using these weights, they cluster the records in the catalog such that all variations are present in a cluster. Our approach on the other hand, mines variational attributes present in the unstructured text in an unsupervised manner, and uses

them to identify both duplicates and variations across two different catalogs. On et al. [2007] links groups of entity variations across catalogs that are the same. Our task is different, where the goal is to identify both duplicates and those that are variations of each other.

**Attribute value extraction:** Unsupervised phrase extraction techniques such as Hasan and Ng [2010], El-Kishky et al. [2014] return a ranked list of phrases (see Hasan and Ng [2014] for a survey). However, they do not explicitly extract product attributes. The task of extracting entity attribute values from unstructured text has received significant attention as well [Zheng et al., 2018, Ghani et al., 2006, Petrovski and Bizer, 2017, Ling and Weld, 2012, Putthividhya and Hu, 2011, Kannan et al., 2011, Köpcke et al., 2012, Ristoski et al., 2017]. These approaches can be broadly classified into closed-word approaches, which assume a predefined set of attribute values [Ghani et al., 2006, Ling and Weld, 2012, Putthividhya and Hu, 2011], and open-world approaches which do not make such assumptions [Zheng et al., 2018]. Rule-based and linguistic approaches approaches such as Chiticariu et al. [2010], Nadeau and Sekine [2007], Mikheev et al. [1999] leverage the syntactic structure of the text to extract to attributes. CRF-based systems such as Putthividhya and Hu [2011] make use of seed dictionary to bootstrap the models. Recently, neural network based models that combine LSTMs and CRF have been proposed [Kozareva et al., 2016, Zheng et al., 2018]. However, all these techniques extract attribute values for a pre-defined fixed set of attributes and are supervised in nature.

## 3. Preliminaries

In this section, we introduce necessary terms and notation, followed by the formal definition of three-way entity linkage.

**Base entity and entity variations:** A *base entity* is an abstract, canonical entity associated with a set of attributes called **base attributes**. *Entity variations* are a set of related entities that are instantiations of the same base entity. All entity variations share the same value for the base attributes. Entity variations are also associated with a set of **variational attributes** whose values differ across variations. We denote the function that maps an entity to its base entity by $B$.

**Records and attributes:** A *record* represents a real-world entity in a data source and is associated with a set of attributes called **record attributes**. Records usually have a combination of structured attributes such as brand and price, and unstructured text attributes such as title and description. The attribute values in a record may be missing or incorrect. Not all base and variational attributes are present as structured record attributes. We use $r_i$ to denote a record, $a_m$ to denote the record attribute and $r_i(a_m)$ to denote the value of the $a_m{}^{th}$ attribute of $r_i$. The set of all attributes is given by $\mathcal{A}$. We denote the function that maps a record to the real-world entity it represents by $E$. The set of all records present in a data source is referred to as a **catalog**, and is denoted by $\mathcal{C}$.

Having described the needed terms, we now formally define the problem of three-way entity linkage.

**Definition.** *Given a pair of records $(r_i, r_j)$ from two catalogs $\mathcal{C}_1, \mathcal{C}_2$, the task of **3-way entity linkage** is to identify whether these records correspond to duplicates ($E(r_i) = E(r_j)$), variations ($\{B(E(r_i)) = B(E(r_j)) \wedge E(r_i) \neq E(r_j)\}$) or distinct entities ($\{B(E(r_i)) \neq B(E(r_j)) \wedge E(r_i) \neq E(r_j)\}$).*

---

**Algorithm 1 VarSpot**: An approach to identify contrast features

---

**Input:** Product catalogs $\mathcal{C}_1$, $\mathcal{C}_2$
**Output:** Contrast features with weights **f**
  # Phase 1: Link each catalog to itself to identify potential variations
  $\mathcal{L}_1 \leftarrow \mathbf{Link}(\mathcal{C}_1)$
  $\mathcal{L}_2 \leftarrow \mathbf{Link}(\mathcal{C}_2)$
  # Phase 2: Extract contrast features from identified variations
  $\mathbf{f_1} = \mathbf{ContrastSpot}(\mathcal{L}_1)$
  $\mathbf{f_2} = \mathbf{ContrastSpot}(\mathcal{L}_2)$
  # Aggregate the contrast features from the two catalogs
  $\mathbf{f} = \mathbf{f_1} \cup \mathbf{f_2}$
  **return f**

---

## 4. Contrastive Entity Linkage

In this section, we introduce the notion of contrast features that capture the value of variational attributes, and propose **VarSpot**, a novel algorithm to identify them. We then describe our overall approach, **Contrastive entity linkage**, that uses **VarSpot** to identify both duplicates and variations.

### 4.1 Contrast Features

*Contrast features* are phrases or n-grams that represent values of variational attributes present in unstructured text attributes such as title. For the motivating example in the introduction, the phrase *16 GB* and *8 GB* are contrast features as they correspond to values of the variational attribute *storage size*. Since contrast features are attribute values of all variational attributes, they are more general. While *pack of 2, 75 oz, dark roast* are some examples contrast features in the groceries domain, *radio edit, remix, live* are examples of contrast features in the music domain.

### 4.2 Overview of VarSpot

**VarSpot** is an unsupervised contrast feature extraction algorithm and has two phases. In the first phase **VarSpot** identifies a set of potential entity variations in a single catalog. This phase returns a set $\mathcal{L}$ consisting of entity pairs and their similarity scores. This phase is described in Subsection 4.3. In the second phase, we use the **ContrastSpot** algorithm to extract phrases that distinguish pairs in $\mathcal{L}$. This is described in Subsection 4.4. We finally aggregate the extracted phrases from both catalogs to generate the contrast features. An overview of **VarSpot** is given in Algorithm 1.

### 4.3 Phase 1: Identifying Potential Variations

The first step of the proposed **VarSpot** algorithm identifies potential entity variations by linking the entities in a catalog to itself. This is motivated by two key properties of catalogs. First, catalogs typically contain very few duplicates. Most sites ensure that records in their catalog refer to distinct entities. Second, the records of variations are typically more similar to each other than to records that correspond to entities with different base products. This is due to the presence of many base attributes that are shared across variations. Based on

the key properties mentioned earlier, we expect that most of the pairs with high similarity scores to be variations.

Typically, in order to limit the $O(n^2)$ potential blowup in candidate pairs, a blocking technique such as locality sensitive hashing (LSH) is used to limit the candidate pairs for linkage. For large catalogs, even after performing blocking, we may need to evaluate a large set of candidate pairs. Since entity variations share the same value for base attributes, we only need to consider record pairs that have the same values for these attributes. We block on the structured base attributes such as brand, when present in the catalog, to reduce the number of pairs.

Further, instead of using a sophisticated supervised linkage system that performs linkage using all attributes of the record, we compute a similarity measure $s$ such as Jaccard or cosine similarity on the unstructured attribute from which we need to extract the contrast features. We then return all pairs that have score $s > \theta$.

The algorithm for identifying potential entity variations **Link** is given in Algorithm 2. The algorithm takes as input a catalog $\mathcal{C}$, structured base attributes $a_b$, bucket size threshold $\lambda$ and similarity threshold $\theta$. First, the algorithm buckets records based on the attribute value $a_b$. We then prune buckets larger than the threshold $\lambda$. For each of the remaining buckets, we consider all record pairs, compute similarity scores and output all pairs greater than threshold $\theta$.

### 4.4 Phase 2: Extracting Contrast Features

The second phase of **VarSpot** extracts contrast features from the unstructured attributes of identified potential entity variations. The main idea is to first start with unigrams that are present in one entity but not the other, and grow the phrase by considering adjacent tokens. We continue until we find the largest *significant phrase*, and extract the phrase as a contrast feature. Similar techniques have been proposed to extract phrases in topic modeling [El-Kishky et al., 2014].

*Significant phrases* are n-grams which occur more frequently than expected had they been sampled independently from their constituent sub-phrases. For example the phrase *pack of 2* could arise either because it is a phrase corresponding to a single attribute, or because {*pack of, 2*} are two independent phrases adjacent to each other. To capture this, we define a *significance score* ($sig$) and consider all phrases $c$ with $sig(c) > \alpha$ to be a significant phrase.

The probability of observing a n-gram $c$ can be estimated by $p(c) = \frac{f(c)}{L}$, where $f(c)$ is the observed frequency of $c$ and $L$ is the total number of all n-grams. A *partition* of a n-gram is a set of smaller n-grams that together form the string. For example, the partitions of the n-gram *pack of 2* are {*pack, of 2*},{*pack of, 2*} and {*pack, of, 2*}. Let $c$ be an n-gram, and $\mathbf{c'}$ be a partition of the tokens in the n-gram. The expected frequency of observing n-gram $c$ due to independently sampling its constituent n-grams in $\mathbf{c'}$ is given by $\hat{f}(\mathbf{c'}) = L * \prod_{t \in \mathbf{c'}} p(t)$, where $p(t)$ is the probability of occurrence of the n-gram $t$. For a n-gram $c$ and one of its partition $\mathbf{c'}$, the standard deviation between the observed frequency of $c$ and the expected frequency due to independently sampling its constituent n-grams in $\mathbf{c'}$ is given by $std\_dev(c, \mathbf{c'}) = \frac{f(c) - \hat{f}(\mathbf{c'})}{\sqrt{f(c)}}$. The significance score $sig(c)$ for an n-gram is given by $sig(c) = min_{\mathbf{c'}} \ std\_dev(c, \mathbf{c'})$

**Algorithm 2 Link**: Scalable unsupervised approach to discover variations

---

**Input:** Catalogs $\mathcal{C}$, Base attributes $a_b$, Bucket size threshold $\lambda$, Similarity threshold $\theta$.
**Output:** Set of potential entity variations $\mathcal{L}$
  # Bucket entities based on base attribute $a_b$
  **for** $r_i \in \mathcal{C}$ **do**
    $H_{a_b}[r_i(a_m)] \leftarrow H_{a_b}[r_i(a_m)] \cup r_i$
  **end for**
  # Prune buckets with size $> \lambda$
  **for** $key \in H_{a_b}$ **do**
    **if** $size(H_{a_b}[key]) > \lambda$ **then**
      Prune $H_{a_b}[key]$
    **end if**
  **end for**
  # Compute similarity and generate pairs
  **for** each pair $(r_i, r_j) \in H_{a_b}$ **do**
    **if** $sim(r_i(a_v), r_i(a_v)) > \theta$ **then**
      $\mathcal{L} = \mathcal{L} \cup (s, r_i, r_j)$
    **end if**
  **end for**
  **return** $\mathcal{L}$

---

**Algorithm 3 ContrastSpot**: Extracting contrast features from potential variations

---

**Input:** Variations $\mathcal{L}$, Unstructured attribute $a_v$, significance score threshold $\alpha$, max. length $m$
**Output:** Contrast features with weights $\mathbf{f}$
  # Compute n-gram frequencies
  **for** $(r_i, r_j) \in \mathcal{L}$ **do**
    Extract n-grams $c$ in $r_i(a_v) \triangle r_j(a_v)$
    $Freq[c] \leftarrow Freq[c] + 1$
  **end for**
  # Compute the set of significant phrases
  **for** $c \in$ Freq with length $> 1$ **do**
    **if** $sig(c) > \alpha$ **then**
      $Sig = Sig \cup c$
    **end if**
  **end for**
  # Extract the longest significant phrase
  **for** $(r_i, r_j) \in \mathcal{L}$ **do**
    $G \leftarrow$ Unigrams present in $r_i(a_v) \triangle r_j(a_v)$
    Grow unigrams in $G$ to largest $c \in Sig$
    $\mathbf{f} \leftarrow \mathbf{f} \cup c$
  **end for**
  **return** $\mathbf{f}$

---

    The **ContrastSpot** algorithm to extract the contrast features for the list of potential entity variations $\mathcal{L}$ is given in Algorithm 3. First, for each pair of entities in $\mathcal{L}$, we extract all n-grams of size up to $m$, that are present in the unstructured attribute of one of the entities but not the other. Since variations share the same value for base attributes, these n-grams typically correspond to variational attributes. We compute the frequency of all extracted n-grams from the set of pairs in $\mathcal{L}$. Next, for all n-grams other than unigrams, we compute the significant score, and keep track of all phrases the have a score greater than the threshold $\alpha$. This is the list of significant phrases. While this procedure can be extended to structured attributes they are better handled by the CEL classifier downstream.

## 4.5 Contrastive Entity Linkage

We now describe our overall contrastive entity linkage algorithm (**CEL**) for the task of three-way linkage. First, we extract the set of contrast features from both the catalogs using the **VarSpot** algorithm. Then, for each entity in the catalog, we extract phrases corresponding to contrast features from the unstructured text attributes, such as title. These phrases correspond to the set of variational attribute values. We add these phrases as an additional record attribute. We train a multiclass classifier using the labeled data to classify a pair of records as either duplicate, variation or distinct. To identify the set of duplicates $\mathcal{M}$, and the set of variations $\mathcal{V}$, we first generate the set of potential duplicate and variation pairs from the two catalogs using blocking techniques such as locality sensitive hashing (LSH). We classify these blocked pairs using the trained classifier, and return sets $\mathcal{M}$ and $\mathcal{V}$.

## 5. Experimental Validation

In this section, we perform experimental evaluation to answer the following research questions: *Q1:* Do the extracted contrast features capture variational attributes? *Q2:* How does contrastive entity linkage perform on the task of identifying variations and duplicates? *Q3:* Does adding contrast features improve the performance of traditional entity linkage frameworks when identifying duplicates?

**Data:** We perform experiments on data of varying sizes from three different product domains: *software, music* and *groceries*. The dataset statistics are given in Table 2.

**Software** is a benchmark e-commerce entity linkage dataset that contains software products extracted from two websites, Amazon and Google [Köpcke et al., 2010]. Each product is associated with three attributes: title, manufacturer and price. We use the same set of blocked pairs as used in Konda et al. [2016]. This is a small-sized dataset with a few thousand entities in each of the catalogs.

**Groceries** contains grocery products sampled from Amazon and products contained in the Open Grocery Database[1]. Each product is associated with two attributes: title and brand. To generate the set of candidate pairs, we performed blocking using locality sensitive hashing (LSH). We used both the attributes for blocking. This is a medium-sized dataset with ~1 million entities in the Amazon catalog and ~100,000 entities in the Opengrocery catalog.

**Music** is a dataset containing music tracks extracted from two different music catalogs, Musicbrainz[Swartz, 2002] and Lastfm[McFee et al., 2012]. Each track is associated with two attributes: title and artist. To generate the set of candidate pairs, we performed blocking using LSH. We used both the attributes for blocking. This is a large-sized dataset with ~1.5 million entities in the Musizbrains catalog and ~1 million entities in the Lastfm catalog.

To train and evaluate the various approaches, we generated a stratified sample of the blocked pairs [Bickel et al., 2009] using the Jaccard similarity computed on the text attributes, and labeled the pairs as duplicates, variations, or distinct. The label distribution for each dataset is given in Table 2. Using the labeled samples, we performed Monte Carlo cross validation, and generated 10 splits by randomly splitting them into train and test splits. We used 70% of the samples for training and use the remaining for testing.

**Entity linkage frameworks:** To evaluate the performance of CEL we make use of Magellan[Konda et al., 2016], a state-of-the-art entity linkage framework. Further, we investigate the utility of the **VarSpot** algorithm for identifying duplicates by integrating the contrast features into three state-of-the-art entity linkage frameworks, SILK [Isele et al., 2010], Magellan and Deepmatcher [Mudgal et al., 2018]. We first extract phrases corresponding contrast features from the title and add it as a separate attribute. For each of the frameworks, we train a model using the augmented data, and compare its performance with a model trained on data without contrast features.

---

1. http://www.grocery.com/open-grocery-database-project/

| Domain | Catalog 1 | Catalog 2 | Blocked pairs | Sampled Duplicates | Sampled Variations | Sampled Distinct |
|---|---|---|---|---|---|---|
| Software | 1364 | 3227 | 11461 | 516 | 564 | 604 |
| Groceries | 1125952 | 110437 | 655254 | 598 | 1215 | 1412 |
| Music | 1456963 | 943335 | 1797549 | 412 | 271 | 1919 |

Table 2: **Data statistics:** Number of entities in each catalog, blocked pairs and label distribution for the three domains.

| | Correct | Partial | Incorrect |
|---|---|---|---|
| **Contrast Features** | **482** | 297 | 436 |
| **Frequent phrases** | 396 | 455 | 364 |
| **Infrequent phrases** | 0 | 114 | 1101 |

Table 3: **Interpretability of contrast features:** We observe that contrast features explain about 40% of variations correctly. Frequent phrases only explain variations partially.

**Magellan** is a entity linkage framework that automatically generates several similarity and distance measures for each of the attributes in the catalog based on the attribute type. Using these measures as features, Magellan trains supervised classifiers such as logistic regression and random forest to perform linkage.

**SILK** is a rule-based entity linkage framework that outputs a set of linked entities across two catalogs. The user specifies the set of attributes to use, similarity measures to compute on these attributes and weights for each attributes. In our experiments, we tuned the weights by performing a grid search using the training data.

**Deepmatcher** is a state-of-the-art deep learning entity linkage framework. It uses word embeddings to generate embeddings of attributes and computes a similarity representation for each of these attributes. Then a classifier is trained on the similarity representation to identify duplicates. We trained a *hybrid* model that uses a bidirectional RNN with decomposable attention and a vector concatenation augmented with element-wise absolute difference to learn a similarity representation.

**Performance metrics:** We compute the F1 score and average precision score (APS) for each of the splits and report the mean and standard deviation. We use the Python *scikit-learn* library to compute the metrics. We performed paired t-test and numbers in bold are statistically significant with $p < 0.05$.

### 5.1 Experimental Results

**Effectiveness of contrast features:** We first provide a qualitative analysis of the extracted contrast features and their effectiveness in capturing the variational attributes. The first step of **VarSpot** identifies potential entity variations by linking the entities in a catalog to itself. For the Link algorithm (Algorithm 2), we used manufacturer, brand and artist as the blocking attributes for software, groceries and music respectively. We pruned blocks greater than 25 for music (large dataset), 50 for groceries (medium dataset), and 100 for software (small dataset). Table 4 shows some sample pairs discovered by the **VarSpot** algorithm for the three domains. We observe that these entity pairs indeed correspond

| Software |
|---|
| peachtree by sage premium accounting for nonprofits 2007 |
| peachtree by sage premium accounting 2007 accountants ' edition |
| peachtree by sage pro accounting 2007 |

| Groceries |
|---|
| milk duds candy 1.85 ounce boxes pack of 24 |
| milk duds candy 5 ounce boxes pack of 3 |
| milk duds movie size 5 oz 12 count |

| Music |
|---|
| groove is in the heart |
| groove is in the heart club version |
| groove is in the heart sampladelic remix |

Table 4: **Identifying entity variations:** Examples of the variations of entities linked by **VarSpot** algorithm for software, groceries and music domains. In these examples, edition is different for software, pack size is different for groceries, and versions are different for music.

to variations. While editions differ for software and pack size for groceries, for music the entities differ on versions.

The second phase of **VarSpot** extracts contrast features from the unstructured attributes of identified variations. We do this by extracting the largest significant phrase present in one attribute but not the other. As most variational attributes consist of 3 or lesser tokens, we extracted phrases up to length 3. We set $\alpha$ to 3.0 and $\theta$ to 0.6. For each of the datasets we extracted the top 100 contrast features by weight using the **VarSpot** algorithm. Table 5 shows some of the top contrast features by weight for different domains. We observe that edition and platform are important for software, package size and flavor are an important for groceries. For music the top contrast features correspond to different versions of the track.

| Software | Groceries | Music |
|---|---|---|
| standard mac upgrade | pack of 6 | remix |
| small box | pack of 2 | mix |
| premium upsell mac | pack of 3 | radio edit |
| standard upsell mac | 2 pack | live |
| deluxe | original | club mix |
| pro | red | instrumental |
| upgrade | strawberry | original version |
| professional | orange | extended mix |
| mac | lemon | acoustic |
| home | premium | part 2 |

Table 5: **Extracted contrast features:** Extracted contrast features for the three domains. Most correspond to variational attributes.

To evaluate Q1 further, we ran annotation tasks using Amazon Mechanical Turk. We provide each worker with a pair of entities that are variations of each other. Along with the entities, we also provide the set of phrases present in the title of one entity but not the other. The worker annotates sets of phrases as correct if they fully explain the reason for products being variations of each other. If not all phrases have been extracted, or only part

| | | | Logistic Regression | | Random Forests | | LR Improv. | RF Improv. |
|---|---|---|---|---|---|---|---|---|
| | | | **NoCF** | **CEL** | **NoCF** | **CEL** | | |
| Software | Duplicates | F1 | 0.753 | **0.784** | 0.785 | **0.81** | 3.1 % | 2.5 % |
| | | APS | 0.832 | **0.864** | 0.877 | **0.897** | 3.2% | 2 % |
| | Variations | F1 | 0.425 | **0.535** | 0.677 | **0.695** | 11 % | 1.8 % |
| | | APS | 0.56 | **0.633** | 0.761 | **0.777** | 7.3 % | 1.6 % |
| Grocery | Duplicates | F1 | 0.658 | **0.706** | 0.717 | **0.741** | 4.8 % | 2.4 % |
| | | APS | 0.72 | **0.767** | 0.809 | **0.835** | 4.7 % | 2.6 % |
| | Variations | F1 | 0.736 | 0.737 | 0.778 | **0.792** | 0.1 % | 1.4 % |
| | | APS | 0.761 | **0.789** | 0.855 | **0.868** | 2.8 % | 1.3 % |
| Music | Duplicates | F1 | 0.665 | **0.703** | 0.781 | **0.793** | 3.8 % | 1.2 % |
| | | APS | 0.747 | **0.782** | 0.854 | **0.869** | 3.5 % | 1.5 % |
| | Variations | F1 | 0.663 | **0.74** | 0.765 | **0.787** | 7.7 % | 2.2 % |
| | | APS | 0.709 | **0.803** | 0.838 | **0.887** | 9.4 % | 4.9 % |

Table 6: **Contrast features improve performance: CEL** significantly outperform models without contrast features (**NoCF**) for both tasks across domains.

of the phrase has been extracted, the worker annotates the set of phrases as partial. If the incorrect phrase or no phrase is extracted, the worker annotates it as incorrect.

We ran the annotation task for the groceries data set as it has the largest set of variations. To show that variational attribute values cannot be captured using just the frequency of phrases, we ran the task with a set of 100 most frequent phrases and 100 most infrequent phrases. These phrases were extracted using the same significance score tests, but by using catalog-wide n-gram frequencies. The results of the experiment are shown in Table 3. We observe that just the top 100 contrast features correctly explain about 40% of the variations correctly. This is 20% more than those explained by frequent phrases. Further, another 20% of the variations are explained partially by the top 100 contrast features. Frequent phrases explain a large number of variations partially as '*of 6*' and '*natural*' are also frequent. Infrequent phrases correspond to rare product-specific attributes perform poorly.

**Performance of CEL:** We evaluate research question Q2 by training logistic regression and random forest models. For the random forest classifier, we set the depth of the tree to 15 and number of trees to 1000. The mean F1 and APS scores for **CEL** and models without contrast features (**NoCF**) are given in Table 6. We observe **CEL** significantly outperforms **NoCF** on all three domains for both duplicate and variation detection. For duplicate detection, the performance boost is up to 4.8% F1 Score and 4.7% APS (logistic regression). For variation detection, the performance boost is up to 11% F1 Score and 7.3% APS (logistic regression). Among the domains, we see significant improvements in the metrics for identifying duplicates for the groceries domain. This is because a large number of variations, which were getting linked as duplicates in **NoCF**, are prevented by the extracted contrast feature attribute.

**Performance on duplicate detection:** To evalute Q3, we train models using the attributes present in the dataset (**NoCF** models) and compare it with models that include contrast features as separate attribute (**CF** models). We train models using Deepmatcher, Magellan and SILK. Since SILK returns the set of linked entities without scores assigned to them, we only report the F1 score for SILK. The mean F1 and APS scores for the **NoCF** and **CF** models are given in Table 7. As in the three-class setting, we observe that **CF** models significantly outperform **NoCF** models on all three domains. For duplicate detection, **CF** models of SILK and Magellan significantly outperform **NoCF** models by

| | | SILK | | Logistic Regression | | Random Forests | | Deepmatcher | |
|---|---|---|---|---|---|---|---|---|---|
| | | **NoCF** | **CF** | **NoCF** | **CF** | **No CF** | **CF** | **NoCF** | **CF** |
| Software | F1 | 0.702 | **0.747** | 0.743 | **0.785** | 0.785 | **0.808** | 0.721 | 0.744 |
| | APS | | | 0.843 | **0.88** | 0.878 | **0.897** | 0.749 | 0.779 |
| Grocery | F1 | 0.614 | **0.629** | 0.681 | **0.722** | 0.708 | **0.735** | 0.647 | 0.674 |
| | APS | | | 0.725 | **0.771** | 0.805 | **0.831** | 0.664 | **0.706** |
| Music | F1 | 0.572 | **0.617** | 0.657 | **0.704** | 0.771 | **0.782** | 0.754 | 0.753 |
| | APS | | | 0.748 | **0.774** | 0.848 | **0.866** | 0.796 | 0.804 |

Table 7: F1 score and APS for the task of identifying duplicates. Models with contrast features (CF) outperform models without contrast features across domains.

up to 4.7% F1 Score (logistic regression). For Deepmatcher, we observe that **CF** models outperform **NoCF** models. However, we also observed a large variance in the metrics across the splits. The random forest models outperform Deepmatcher on all the tasks. Similarly, logistic regression outperforms Deepmatcher for the software and grocery domains. This might be due to the small amount of training data and large number of parameters present in the Deepmatcher model. For example, for the software domain, Deepmatcher models have ∼7.1 million parameters for the **NoCF** models and ∼9.2 million parameters for the **CF** models. Mudgal et al. [2018] make similar observations and show that Magellan models outperform Deepmatcher models when the number of labeled data is less than 10,000.

## 6. Conclusion and Future Work

In this paper, we proposed our approach, contrastive entity linkage, to identify both entity pairs that are duplicates and those that are variations of each other. To address the challenge of identifying variational attributes present in the unstructured text, we proposed a scalable, unsupervised algorithm, **VarSpot**. In the experiments, using Mechanical Turk, we first showed that the contrast features are interpretable and then showed that adding contrast features as a separate attribute, outperforms three state-of-the-art entity linkage systems.

This work suggests other interesting future directions. The distinction between variations and exchangeable products can often be subjective. For some consumers, the distinction between a low-sodium versus regular soup is irrelevant; for others it is highly important. An interesting direction for further work is developing and testing algorithms for personalized entity linkage. Another direction of future work is in identifying the type of product attribute captured by the contrast feature. This can enable a more fine-grained discovery of product variations such as variations that differ on flavor, variations that differ on package size and so on.

## 7. Acknowledgements

This work was partially supported by the National Science Foundation grants CCF-1740850, IIS-1703331, the Defense Advanced Research Projects Agency and an Amazon Research Award.

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

## Appendix A. Analysis of Entity Linkage

To analyze the performance of models using contrast feature we show the confusion matrix for one of the folds in the Music dataset in Table 8. We observe that adding contrast features helps the random forest model identify duplicates correctly which were earlier getting classified as variations. As an example the track "harvest uptown famine downtown" in catalog 1 and the track "harvest uptown" in catalog 2 was classified as a variation by the model without contrast features. This is likely because the similarity measures between the two tracks are close to that of variations. However, this was correctly classified as a duplicate by the model with contrast features as "famine downtown" in not a variational attribute and was not extracted as a contrast feature.

Without contrast features

|  | Pred Distinct | Pred. Dup. | Pred. Var. |
|---|---|---|---|
| Distinct | 570 | 8 | 4 |
| Dup. | 11 | 83 | 23 |
| Var. | 7 | 7 | 68 |

With contrast features

|  | Pred Distinct | Pred. Dup. | Pred. Var. |
|---|---|---|---|
| Distinct | 570 | 8 | 4 |
| Dup. | 13 | 91 | 13 |
| Var. | 7 | 6 | 69 |

Table 8: **Confusion matrix:** Models with contrast features correctly identify duplicates which are classified as variations by models without contrast features.

## Appendix B. Hyperparameter Tuning

The **VarSpot** algorithm and Contrastive entity linkage has four main hyperparameters - the significance threshold $\alpha$, the bucket size $\lambda$, the similarity threshold $\theta$ and the number of top contrast features used. In our experiments we set $\alpha = 3$, $\theta = 0.6$ and considered the top 100 contrast features by weight. Further we set $\lambda = 25$ for the music domain. Fig. 2 shows the sensitivities of these parameters to the entity linkage and variant linkage metrics (F1, APS). We plot the metrics as we vary the parameters individually. We observe that **VarSpot** algorithm and Contrastive entity linkage is robust to changes in these parameters.

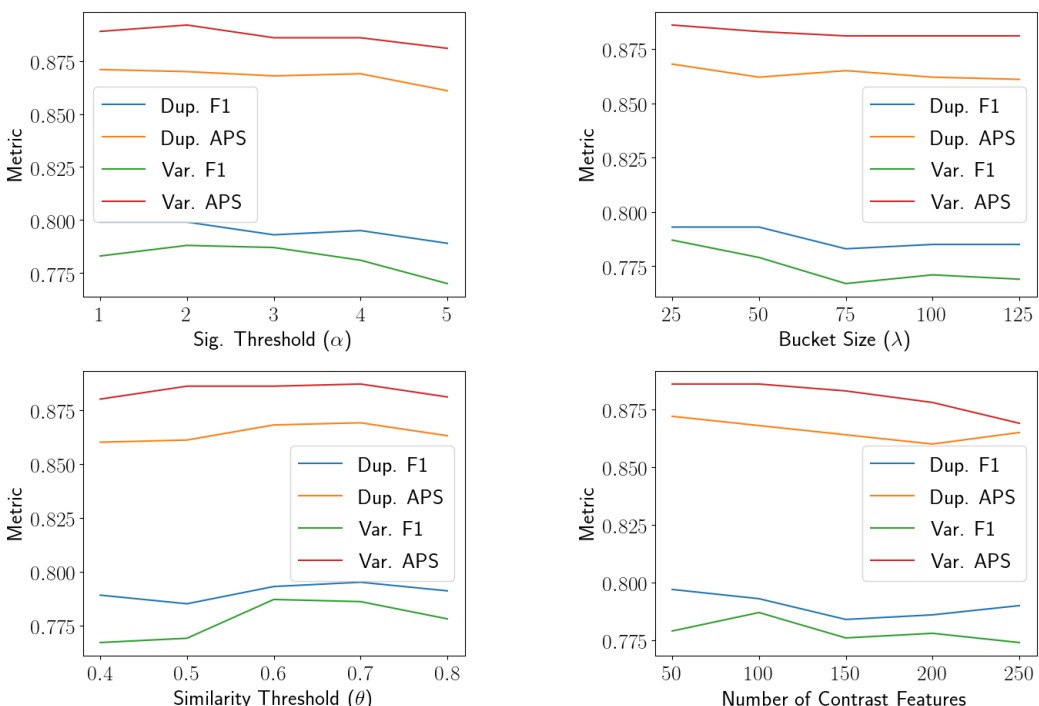

Figure 2: **Sensitivity Analysis:** Plot of entity linkage and variant linkage metrics as we vary the hyperparameters. Our approach is robust to changes in these hyperparameters.