# OpenReview forum: "Contrastive Entity Linkage: Mining Variational Attributes from Large Catalogs for Entity Linkage"
_AKBC.ws/2020/Conference — AKBC 2020_

### Official Review · AnonReviewer1 · 2020-03-26
**nice paper but maybe limited to product databases**

**Rating:** 7
**Confidence:** 4

**Review:**

Update after response:

Thanks to the authors for responding to my comments in particular adding the parameter analysis. While I still wonder about the limited scope of the solution, the research is nicely done and the problem formulation is now clearer.

-------------
Pros:
- novel problem definition
- experiments on multiple datasets
- nice usage of feature extractor in multiple

Negatives
- problem formulation does not include some assumptions
- potential lack of generalizability

This paper describes the problem of entity resolution in an environment where there are a wide variety of entity variations. I thought this was a rather novel problem formulation. They introduce the notion of contrastive entity linking to solve this problem. In particular, they define a blocking mechanism for identifying entity variations and a feature extraction algorithm for identifying entity attributes that are core to the entity or that are part of the entity's variation. These can then be used to drive a classifier.

My main criticism of the paper is the potential generalizability. While it's applied in three different domains, the datasets essentially of the same kind, namely, product databases which already contain unique entities. From my reading, the assumption is not stated in the problem definition.

The problem definition could be more precisely worded. In section 3.3, two assumptions are stated about catalogs, namely, that they ensure that records refer to distinct entities and that entity variation (i.e. record variations) are more similar to each other than base entities. These are important assumptions that make the problem much easier than what was outlined in the problem definition.

In terms of evaluation, the paper didn't seem to report a number of critical parameters, namely , the bucket size threshold and similarity threshold during the experiments.

I appreciated the experimental settings of using the feature extractor in a number of downstream entity.

There's a couple pieces of related work. First, for entity resolution I think this approach bears similarity to [1]. There's been quite a bit of work in the NLP community on identity (see e.g. [2]) that would be useful to discuss.

Overall, I thought the paper was a nice contribution.

Minor comments:
- The paper was easy to read.
- it would be good to check the usage of the word record, entity and product, they get confused in places.
- It would be nice if the annotated data is also made available in the paper.


[1] Zhu, Linhong, Majid Ghasemi-Gol, Pedro Szekely, Aram Galstyan, and Craig A. Knoblock. "Unsupervised entity resolution on multi-type graphs." In International semantic web conference, pp. 649-667. Springer, Cham, 2016.

[2] Recasens, Marta, Eduard Hovy, and M. Antònia Martí. "Identity, non-identity, and near-identity: Addressing the complexity of coreference." Lingua 121.6 (2011): 1138-1152.

---

> ### Author Response · Authors · 2020-04-10
> **Updated the draft to address reviewer's comments**
>
> We thank the reviewer for the time and helpful feedback.
>
> As previously mentioned, we have updated the draft of paper with values of all hyperparameters and have added an appendix section containing sensitivity analysis for all the parameters. We show that the approach is robust to a wide range of parameter values. We have also included a list of top extracted contrast features for each dataset.
>
> We have also added a related work to include paper pointed out by the reviewer, thank you for these suggestions.
>
> With regards to generality, contrast features can be extracted and used for entity resolution in domains such as health records (potential contrast features can be Jr., Sr., II etc.). However, online retail is an important domain and handling variations during entity linkage, while crucial,  is still a challenge. Our experiments on products show how that our approach can address this challenge.
>
> We have reworded the problem definition to make it easier to understand.
>
> In section 3.3, we make an observation that the variations are similar as they share the same values for all base attributes, and catalogs typically do not contain a large number of duplicates. This motivates out two phase extraction.  Our approach is applicable even when the catalogs contain duplicates (In fact all our datasets do contain duplicates).

---

### Official Review · AnonReviewer3 · 2020-03-28
**the paper an interesting work in progress which requires improvements in (a) novelty (b) analysis.**

**Rating:** 7
**Confidence:** 4

**Review:**

Paper is about finding variational attributes for catalogue named entities.  Examples of such attributes is "capacity" for a memory card (e.g. Sandisk flash drive "64GB") and identification of such attributes helps in duplicate detection in E-commerce cataloging and search.  The proposed approach is unsupervised where they first detect some candidate entity variations (pairs of entities with high similarity scores) and then in each pair detect the "significant" phrase that "contrasts" one entity from the other one in the pair as the contrastive feature.  The significance phrases (ngrams) is estimated exhaustively from a corpora with a PMI-like metric.  Authors experiment with three entity linking systems with diverse architecture ranging from rule-based to logistic regression and neural-based models on three domains (music, grocessary and software catalogs).  Results are promising and show that most systems benefit from these features.

Paper is written mostly well: problem has been defined and motivated well and the approach is presented in a smooth structure and flow.  However, presentation of results and analysis is unclear at parts.  Novelty of the approach is modest and is mostly around the detection of detection of contrast features and significant phrases.  These approaches which show promising improvements against the baseline, can be expanded to more recent efforts in using deep semantic representations in NER and extraction of multi-word-expression.  Experiments are fairly extensive and support the proposed approach well.  Analysis is not extensive and should be improved.  All together, I found the paper an interesting work in progress which requires improvements in (a) novelty (b) analysis.

Questions and suggestion:

1. Evaluation of candidate pairs is based on a data that is annotated in a post-extraction fashion (annotator labels the output of the system).  So if you don't have a gold-standard set (all possible pairs), how do you compute "recall" there? (to compute the F score in table 4).

2.  Did you experiment with richer models of semantic similarity using embedding, etc?

3. Despite the preceding explanation of the notation elements, the formal definition of the 3-way entity linkage is not easy to understand and doesn't connect with the rest of the section.

4. In the core extraction of significant phrases and contrast features, ngram frequency is the major factor (along with some thresholding).  It is not clear why authors are comparing their interpretability against "frequent phrases" (which are a fairly similar approach).  Please elaborate more on Table 3 comparison.

 5. Please provide more details and analysis on results of the three way classification, specially around the confusion metrics.  Are the improvements similar for different classes.  What kind of duplicates does the CF model extract that the No-CFs don't, etc.

6. How would analyze last column of table 3 (higher rate of incorrect class for contrast features)?

Post Rebuttal comment:
After reading authors responses to my and other reviwers' comments and also checking the new draft, I am going to lift my rating of the paper.  Thanks for your willingness to improve your work.

---

> ### Author Response · Authors · 2020-04-10
> **Updated the draft to address reviewer's comments**
>
> We thank the reviewer for their time and helpful feedback.
>
> As mentioned in the previous response, we have updated the draft of paper with values of all hyperparameters and have added an appendix section containing sensitivity analysis for all the parameters. We show that the approach is robust to a wide range of parameter values. We have also included a list of top extracted contrast features for each dataset.
>
> Question & suggestions
> 1)For evaluation, we created a stratified sample of the blocked pairs using the Jaccard similarity computed on the text attributes, and labeled the pairs as duplicates, variations, or distinct. This set is independent of the system. We predict using the various trained approaches and compute the metrics.
>
> 2)We do use semantic similarity measures such as embeddings in the classifiers that perform the 3-way classification. The deep learning based DeepMatcher model has an embedding layer that uses the embeddings of contrast features. This layer should help the model generalize contrast features. We have not tried using embeddings in the contrast feature extraction phase and it is an interesting future direction.
>
> 3) We have updated the problem definition which makes it easier to understand.
>
> 4) While the contrast features look at different significant phrases present across variations, frequent phrases are phrases that occur frequently in a catalog. For example, “pack of 2” is a frequent significant phrase. This is the reason why 30% of variations are explained by frequent phrases. However, not all frequent phrases are variational attributes (e.g., famous brands such as Betty Crocker). As a result contrast features can explain more variations.
>
> 5)We have included the confusion matrix and its analysis in the appendix section.
>
> 6) We used the top 100 contrast features for the entity resolution experiments. For consistency, we used the same top 100 contrast features and could explain about 60% of the pairs (either partially or fully). We could potentially explain a lot more variations with a large set of contrast features.
> However, some variations  are very specific to the product (E.g. huggies pull-ups disney learning designs training pants size 2t-3t, huggies pull-ups disney learning designs training pants size 4t-5t) and contrast features cannot explain them.

---

### Official Review · AnonReviewer2 · 2020-03-30
**Mining significant phrases to identify variants of entities to improve entity resolution**

**Rating:** 5
**Confidence:** 4

**Review:**

The authors propose a new algorithm, contrastive entity linkage (CEL), to identify duplicates and variations of entities in catalogues. The authors introduce the concept of base entities and entity variations. A base entity is defined using a set of attributes, and all variations must have the same values for base attributes, but differ in the non-base attributes. The key idea is to mine significant phrases from the unstructured attributes of a record such as the title or a product. The significant phrases are added as a new attribute, and a classifier is trained using the new "variational" attribute. Experiments show that inclusion of the variational attribute improves entity resolution results.

pros:
- the work is in an important area as entity resolution of near duplicates remains a challenging task.
- method is unsupervised so can be easily applied in new domains
- method improves the performance of any ER system (as it defines a new feature)

cons:
- distinction of base and variational attributes is unclear in practice (see below)
- no discussion of hyper-parameter tuning
- hard to replicate, no reference to open code, several details not fully specified

The key contribution of the paper is the VarSpot algorithm to identify variational attributes (contrast features). The main idea is to mine word ngrams whose frequency is larger than expected based on the frequency of the individual parts of the ngram. This idea is similar to the significant terms query in ElasticSearch.

The evaluation focuses on three datasets, Amazon/Google software products, groceries, and musicbrainz/lastfm. The evaluations show that the contrast features improve the entity resolution performance on all three datasets for identification of duplicates and variants. The evaluation compares results with and without the contrast features, showing that the three ER systems considered in the evaluation (SILK, Magellan and DeepMatcher) benefit from the contrast features. In all experiments random forest consistently outperforms logistic regression so nit doesn't seem useful to include both.

The algorithm has two hyper-parameters, the threshold alpha to prune the significance of an ngram, and the length of the ngrams. The paper does not discuss how these hyper parameters were optimized, or sensitivity to them.

The distinction between base and variational attributes is unclear. In many cases, unstructured fields such as titles or descriptions may include both base and variational attributes (how are they distinguished?). Also, variational attributes may appear in structured fields too (eg memory size can be a structured attribute). In these cases it is unclear how the ngrams are identified. This part should be made clearer in the paper.

In summary, the paper presents an interest variant of an old problem, and presents a simple method to extract a useful feature from the unstructured attributes in records. The evaluation shows promising results, but is not thorough as it should evaluate the hyper-parameters that are used in constructing the feature. The paper is clearly written and accessible to a wide audience. The related work is incomplete as there isn't a related work section, or a discussion of relevant work on mining of significant phrases.

---

> ### Author Response · Authors · 2020-04-10
> **Updated the draft to address reviewer's comment**
>
> We thank the reviewer for the time and helpful comments.
>
> We have updated the draft of paper with values of all hyperparameters. With these values the model is fully specified and should be reproducible. We have also added an appendix section containing sensitivity analysis for all the parameters. We show that the approach is robust to a wide range of parameter values.
> The reviewer rightly points out that fields such as title contain both base and variational attributes . In our approach, we only extract significant phrases present in one of the entities but not the other. Since base attributes are common to all entity variations the extracted phrases mostly correspond to variational attributes. We have included the top extracted phrases in the appendix and which substantiates the claim.
>
> Further, as pointed out by the reviewer, variational attributes can also be present as structured attributes. This can be easily handled by the classifier that performs EL and we do not extract them.
>
> We have clarified these both points in the paper and have also added the related work to include significant phrase mining approaches .

---

### Decision · Program_Chairs · 2020-05-01

**Decision:**

Accept

**Comment:**

This paper addresses the problem of unsupervised duplicate resolution of attributes for e-commerce and propose a new approach for this, which they call "contrastive entity linking". Overall, the reviewers agree that the paper deals with an important problem, and that it is well-written and motivated.